# Immunothrombotic Mechanisms Induced by Ingenol Mebutate Lead to Rapid Necrosis and Clearance of Anogenital Warts

**DOI:** 10.3390/ijms232113377

**Published:** 2022-11-02

**Authors:** Stephan A. Braun, Alexander T. Bauer, Csongor Németh, Annamária Rózsa, Louisa Rusch, Luise Erpenbeck, Sebastian Schloer, Steffi Silling, Dieter Metze, Peter A. Gerber, Stefan W. Schneider, Rolland Gyulai, Bernhard Homey

**Affiliations:** 1Department of Dermatology, University Hospital Muenster, 48149 Muenster, Germany; 2Department of Dermatology, Medical Faculty, Heinrich-Heine University, 40225 Duesseldorf, Germany; 3Department of Dermatology and Venereology, University Medical Center Hamburg-Eppendorf, 20251 Hamburg, Germany; 4Department of Dermatology, Venereology and Oncodermatology, University of Pécs, Medical Center, 7632 Pécs, Hungary; 5Department of Dermatology, Venereology and Allergology, University Medical Center Göttingen, 37075 Göttingen, Germany; 6Center for Molecular Biology of Inflammation, Institute of Medical Biochemistry, University of Muenster, 48149 Muenster, Germany; 7Leibniz Institute of Virology, 20251 Hamburg, Germany; 8Institute of Virology, National Reference Center for Papilloma- and Polyomaviruses, Faculty of Medicine and University Hospital Cologne, 50935 Cologne, Germany

**Keywords:** immunothrombosis, neutrophil extracellular traps, human papillomavirus, thrombosis, von Willebrand factor

## Abstract

Ingenol mebutate (IM) is highly effective in the treatment of human papillomavirus (HPV)-induced anogenital warts (AGW) leading to fast ablation within hours. However, the exact mode of action is still largely unknown. We performed dermoscopy, in vivo confocal microscopy (CLM), histology, immunohistochemistry, and immunofluorescence to gain insights in mechanisms of IM treatment in AGW. In addition, we used in vitro assays (ELISA, HPV-transfection models) to further investigate in vivo findings. IM treatment leads to a strong recruitment of neutrophils with thrombosis of small skin vessels within 8 h, in a sense of immunothrombosis. In vivo and in vitro analyses showed that IM supports a prothrombotic environment by endothelial cell activation and von Willebrand factor (VWF) secretion, in addition to induction of neutrophil extracellular traps (NETosis). IM superinduces CXCL8/IL-8 expression in HPV-E6/E7 transfected HaCaT cells when compared to non-infected keratinocytes. Rapid ablation of warts after IM treatment can be well explained by the observed immunothrombosis. This new mechanism has so far only been observed in HPV-induced lesions and is completely different from the mechanisms we see in the treatment of transformed keratinocytes in actinic keratosis. Our initial findings indicate an HPV-specific effect, which could be also of interest for the treatment of other HPV-induced lesions. Larger studies are now needed to further investigate the potential of IM in different HPV tumors.

## 1. Introduction

Ingenol mebutate (IM), a first-in-class macrocyclic diterpene ester, is effective in the topical field therapy of actinic keratosis (AK) [1,2]. However, in the last few years, it has been proven that IM is also effective in the treatment of various other skin diseases [3]. The most impressive results are certainly seen in the treatment of anogenital warts (AGW). AGW are benign epithelial tumors caused by an infection with certain types of human papillomavirus (HPV) [4]. Many individual case reports, smaller case series, and initial pilot studies have shown, that IM gel application effectively and quickly removes AGW [5,6,7,8,9,10,11,12,13,14]. In most cases, a single application of the gel was sufficient to ablate the warts within 24 h. After 8 h, the treated warts showed clinical signs of necrosis and peeled off within the next 24 h. Some patients felt severe pain at the treatment site, which peaked after 8 h and quickly ameliorated over the next half day. This clinical course is completely different from the reaction that IM induces in the treatment of AK. Here, patients gradually develop erosive, pustular dermatitis after two or three applications on consecutive days [2]. The dermatitis peaks after 4 days and usually needs up to 4 weeks to resolve completely [2]. The different clinical courses indicate that the mode of action (MOA) of IM in the treatment of AGW fundamentally differs from that of AK.

The underlying mechanism of biological effects following topical IM treatment are largely unknown and may rely on the pathophysiological context. The understanding, to date, is based primarily on in vitro studies in tumor cells and a few in vivo studies in murine tumor models [15]. Only one study on the MOA in AK was conducted on human tissue [16]. According to the current understanding, the anti-tumor effects of IM are primarily based on a dual MOA [15]. At high concentrations (>100 µM), which are only attained in the epidermis when applied topically, IM induces direct tumor cell death [17,18,19]. The anti-tumor effect of low doses of IM (~10^−7^ M) is mainly mediated by the activation of protein kinase C (PKC) and the PKC-dependent release of proinflammatory cytokines and chemokines, and thus, the activation of endothelial and immune cells [20,21]. Both the activation of the endothelium and the proinflammatory response results in the recruitment of a neutrophil-rich inflammatory infiltrate inducing an anti-tumor immune response [16,22,23,24,25,26].

So far, very little is known about the MOA of IM in the treatment of HPV-induced tumors, such as AGW. The first ex vivo studies have shown that the impressive rapid necrosis of AGW is not induced just by simple cytotoxic effects [27]. In this study, we further investigated the MOA of IM, focusing on the underlying mechanisms, which lead to fast ablative necrosis during the IM treatment of AGW.

## 2. Results

### 2.1. IM Induces Thrombosis in Capillaries in AGW

To gain further insights into the clinical response of AGW to IM, we performed dermoscopy and in vivo confocal laser microscopy (CLM) on a sample patient. The patient had one single wart at the urethral orifice (Figure 1a). We did dermoscopy at 8 h post treatment. The dermoscopy of the wart’s vessels revealed a partly livid erythematous color with distinct caliber variations and large areas of necrotic epithelium (Figure 1b, t8). We further observed in video CLM that the blood flow in many capillaries of the papillary body completely stopped after 8 h, indicating microthrombosis (Figure 1c, Appendix A).

### 2.2. Histopathology Confirmed Microthrombosis with Neutrophils

We performed histologic examination on the IM-treated warts of seven patients. We focused on criteria for epidermal cytotoxicity, vasculitis, and vasculopathy (Table 1). All warts showed histologic criteria for toxic effects at the epidermis. After 3 h, the epidermis had paled, and after 8 h, it showed ballooning of keratinocytes and reticular degeneration. Histologically, we could also confirm thrombosis in the capillaries of the dermal papillae in five out of our seven patients after 8 h. We detected fibrinoid thrombi (Figure 2a) in capillaries surrounded by sludge of erythrocytes. Patient 7 already showed a thrombus after 3 h. In three patients, AGW were excised deeper, so that postcapillary venules could be analyzed. The postcapillary venules showed a strong diapedesis of neutrophils with extravasated erythrocytes and some leukocytoclasia around the vessels (Figure 2a). However, evidence of vasculitis, such as a clear fibrinoid degenerative changes of the vessels’ walls, were missing. Immunohistochemically, we could demonstrate that the attraction and migration of myeloperoxidase (MPO)-positive neutrophils already started after 3 h. We could detect first perivascular-located neutrophils in most of the cases, sometimes combined with intravascular neutrophils and diapedesis of neutrophils. After 8 h, the inflammatory infiltrate consisted mainly of neutrophils (Figure 2b). In addition, there were abundant CD8-positive cytotoxic T cells, and some CD4-positive T helper cells, and CD68-positive macrophages mixed within the infiltrate (Figure 2b). Due to differences in the excision depth of the shave excisions, not all criteria could be analyzed equally well in all warts. Therefore, due to the limited number of cases, a statistical evaluation did not yield significant results. However, trends were clearly identifiable. The detailed evaluation of the histological and immunohistochemical criteria is presented in Table 1. Additional pictures of two other patients are presented in the Appendix A.

### 2.3. Immunofluorescence Analyses Demonstrated Early VWF Secretion and Formation of NETs

After demonstrating thrombosis with neutrophils by CLM and histology, we further looked for possible thrombotic mechanisms. To investigate the potential role of prothrombotic VWF in IM-treated warts, we evaluated VWF release by immunofluorescence microscopy (Figure 3a). Intraluminal VWF fibers were observed already after 3 h following IM treatment (t3) and were even more pronounced after 8 h (t8). When stretched in the blood flow, luminal VWF fibers bind to platelets [28] and neutrophils [29,30]. We detected a strong correlation between intraluminal VWF fibers and neutrophils. In line with the VWF fiber formation, we could confirm that neutrophil recruitment already started after 3 h. After 8 h, a strong luminal infiltration of CD15+-positive neutrophils was observed (Figure 3a, lower panel left).

Since neutrophils with the formation of NETs is also linked to inflammation-induced thrombosis [31], the presence of NETs was assessed by staining for citrullinated H3 (H3cit, green), neutrophils were stained by antibodies against MPO (red) (Figure 3b). At baseline (t0) and at 3 h after treatment (t3), neither neutrophils nor NETs were visible in the histological sections. However, after 8 h (t8), neutrophils and NETs were clearly visible within the papillary dermis, as exemplified in the immunofluorescence images of patient 6 (Figure 3b). The other patients showed a similar distribution of neutrophils and NETs. The evaluation of two other patients is shown in Appendix A.

### 2.4. In Vitro Analyses Confirmed Endothelial Cell Activation and VWF Secretion after IM Stimulation

Our results demonstrate that VWF fibers in the warts’ blood vessels are associated to increased neutrophil counts and NETosis. Under physiological conditions, VWF is stored in endothelial cells of the vessel wall. Our own studies demonstrate that different stimuli, such as thrombin and VEGF-A, can induce the activation of endothelial cells followed by the exocytosis of VWF [32]. Therefore, to determine if IM cold triggers the secretion of VWF, we used primary human umbilical vein endothelial cells (HUVECs). As shown in Figure 4a, a typical punctate distribution of VWF in Weibel–Palade bodies was observed in the quiescent endothelium. The stimulation of HUVECs with IM induced VWF secretion, reflected by the formation of VWF strings on the luminal surface. ELISA measurements in the supernatant revealed an almost three-fold increased secretion of VWF following stimulation with thrombin. After IM treatment, VWF secretion increased ~three-fold. Of note, the transmembrane endothelial cell marker CD31 was restricted to cell–cell contacts and was distributed similarly in untreated controls and after incubation with thrombin (not shown) and IM (Figure 4a). These data indicate that IM directly triggers endothelial cell activation and VWF release without significant toxic cell damage.

### 2.5. IM Superinduces IL-8 (CXCL8) Expression in HPV-E6/E7 Transfected HaCaT Cells

As the clinical response to IM of HPV-induced AWG fundamentally differs from the response of AKs, we thought that the difference in the response to IM treatment might be due to the presence of the HPV infection of keratinocytes in AGW. We have earlier reported that IM treatment elevates IL-8 expression to a greater extent in HPV-associated epithelial tumor cells than in HPV-negative human epithelial keratinocytes [26]. To mimic HPV infection and recapitulate the relevance of the HPV pathogenic risk factor proteins E6 and E7 in vitro, we used the spontaneously transformed HPV-negative keratinocyte cell line (HaCaT) from histologically normal skin transfected with the control or the HPV16 E6/E7 plasmid and analyzed the effect of E6/E7 expression on IL-8 production. The association between elevated IL-8 concentrations and venous thrombosis is well-known and most likely linked by the IL-8 ability to induce tissue factor, an important inducer of blood coagulation, on monocytes and IL-8’s positive effect on leucocyte recruitment [33,34]. Both can form a microenvironment in which thrombosis is facilitated [35]. We could show that IL-8 expression was six-fold higher in HPV-transfected HaCaT cells when compared to non-transfected controls (Figure 4b). This result indicates that the differential clinical response may be due to high IL-8 expression induced by HPV infection.

## 3. Discussion

The clinical efficacy of IM during the treatment of HPV-induced epidermal proliferation in AGW has been independently reported by several groups. Yet, to date, the underlying mechanisms remained largely elusive. Here, we report for the first time, that IM treatment resulted in thrombosis in the capillaries and postcapillary venules in AGW. While thrombosis has not been described as an MOA of IM, particularly not in the treatment of AK [16], it may explain well the clinical course observed in patients with AGW upon IM treatment. Patients treated with IM reported 8 h later a thrombo-ischemic pain at the treatment site, similar to what is reported in patients with livedoid vasculopathy. The vascular occlusion contributed to rapid necrosis of the epidermis, which finally resulted in the wart’s ablation within 24 h. Interestingly, the ex vivo treatment of warts with IM showed only minor cytotoxic effects on keratinocytes and no rapid necrosis [27]. The non-concurrent effect of IM in the in vivo and ex vivo situation might be explained by the disturbed blood circulation and absence of a neutrophil-driven response directed towards the virus-infected area in the ex vivo scenario. In this in vivo study, we also detected an early fading of keratinocytes as a histological sign of toxic damage of keratinocytes in all warts within the first 3 h, when no significant obliterating vasculopathies were yet present. Cytotoxic concentrations of IM (>100 µM) can be attained in the epidermis after 2 h when IM gel is topically applied [19]. We, therefore, conclude that the additional direct toxic effect of IM also occurs at an early stage and contributes to wart necrosis, but is not sufficient for the fast necrotic detachment. The major MOA for rapid wart necrosis seems to be the immune response inducing thrombosis. We could demonstrate that IM treatment leads to a strong recruitment of a neutrophil-rich infiltrate also in AGW, as it has already been described in both human tissue of AK and in several murine tumor models [16,24,36]. Additionally, we detected significant NET formation. The appearance of NETosis might relay on two cellular IM effects: IM affected the production of neutrophil attractor and inducer of NETosis IL-8 [26,37], and further induced NETosis via PKC-activation [38]. Furthermore, we showed that IM supports the release of stored VWF from endothelial cells. The effect was even higher than that of thrombin, which promotes endothelial cell activation via a protease activated receptor (PAR-1) [32,39]. Neutrophils with NET formation and release of VWF from endothelial cells are strongly implicated in thrombosis, and therefore likely contribute to the vascular alterations which ultimately lead to the detachment of the AGWs [40]. The formation of thrombi is a complex process and is usually multifactorially triggered involving complex prothrombotic and proinflammatory processes [40]. This is in line with our data indicating that there is no single mechanism responsible for thrombosis, but rather a combination and interaction of many different factors. First, exocytosis of VWF allows the formation of long VWF fibers, which contribute to a pro-thrombogenic environment by primarily promoting platelet binding and aggregation [41,42]. In this context, the release of VWF may be directly induced by IM via activation of protein kinase C (PKC) isoforms. IM is an activator of PKC-delta, which is known to mediate VWF secretion from endothelial cells in response to VEGF [23,43]. Second, neutrophils with the formation of NETs are the key player of so called immunothrombosis, which has a major physiological role in immune defense and facilitates the recognition, containment, and destruction of intravascular pathogens [40]. NETs support thrombosis through many pathways: (i) they directly activate factor XII, (ii) bind to VWF and support the recruitment of platelets, (iii) trigger activation of platelets, (iv) induce cleavage of anticoagulants, and (v) bind to tissue factor to promote extrinsic coagulation [40]. In our scenario, we assume that IM, rather than pathogens, directly activates endothelial cells, leading to the recruitment and activation of neutrophils and NETs, and thus, triggers pathways of immune thrombosis in a non-physiological, hyperactivated manner. This conclusion is also supported by the fact that neutrophil-driven immunothrombosis is strongly promoted by IL-8 [44], the key chemokine induced by IM [15]. IM also directly induces the neutrophil respiratory burst with consequent production of reactive oxygen species [25]. Excessive release of NETs and cytotoxic granule contents contributes to tissue damage [45], and could additionally enhance thrombosis. Most recently, it has been impressively demonstrated that neutrophil hyperactivation and NET formation contribute to the occurrence of immunothrombosis in coronavirus disease 2019 (COVID-19) patients [46,47,48,49]. These patients showed microthrombi formation within capillaries and small vessels, similar to the histological changes we observed in the IM-treated warts [50,51].

The finding that microthrombosis is only induced in HPV-induced AGW and absent in AK is of particular interest and may be explained by differences in IL-8 levels. We had previously shown that IM induces IL-8 expression in HPV-associated epithelial tumor cells to a greater extent compared to human epithelial keratinocytes [26]. Here, we showed that keratinocytes expressing HPV proteins E6 and E7 also produce IL-8 to a six-fold greater amount compared to non-HPV protein expressing keratinocytes. Although this experimental approach certainly does not ideally reflect HPV infection in AGW, these data indicate that HPV-altered keratinocytes secrete particularly high amounts of IL-8 following IM stimulation. These levels may be then sufficient to create a microenvironment in which thrombosis is triggered. Moreover, recent transcriptome analyses of common warts have shown that the top 500 differentially expressed genes were strongly associated with neutrophil degranulation [52]. A stronger release of cytotoxic granules from neutrophils in HPV-induced warts may also amplify the thrombogenic effects. Li et al. suggested that IM acts through epidermal multidrug transporter (ABCB1)-mediated absorptive transport, leading to fast dermal penetration and inducing vascular damage [36]. Direct vascular damage could be an additional trigger for the formation of thrombosis in AGW. It has been described that in HPV-induced tumors ABC-transporters are upregulated [52,53]. An efficient transport of IM to dermal vessels in HPV-associated tumors, inducing vascular damage, could thus be another factor initiating and driving thrombosis. It is also known that viruses influence the vascular network of tumors they induce [54]. For example, HPV causes upregulation of VEGF, and thereby promotes angiogenesis and vascular permeability [54]. A different activation status of vessels in HPV-induced tumors could also explain the different response of HPV-positive AGW and HPV-negative AK to IM. We also tested IM on vessels of HPV-negative cherry hemangiomas. In CLM, cherry hemangiomas show comparable vessels and blood flow as seen in AGW. IM did not induce thrombosis in these tumors.

At this point, however, it must also be noted that this study shows limitations. We were able to show thrombosis in AGW in seven patients, but HPV-negative controls, such as surrounding healthy skin, were missing to attain more in vivo evidence for HPV specificity Due to a lack of well-established and easily accessible in vitro assays for low-risk HPV infection, an in vitro proof of HPV-specificity is very difficult. As stated before, our in vitro analysis with immortalized keratinocytes transfected with viral proteins of high-risk HPV did not optimally represent the biology of a low-risk HPV infection. It has also been shown that undifferentiated keratinocytes express more IL-8 on IM stimulus compared to differentiated keratinocytes, so that by just using immortalized HaCat, increased IL-8 expression was expected [16]. Yet, additional expression of viral HPV proteins seems to enhance this effect.

To sum up, it can be noted that there is evidence to suggest that HPV-induced tumors, as well as their microenvironment, are particularly sensitive to IM treatment. An HPV-specific response might be of particular clinical interest, as it could also be used to specifically target vasculature of other, also malignant, HPV-associated tumors, such as head and neck cancer, cervical, vaginal, and anal cancer. Research on larger patient collectives with appropriate controls is now needed to further investigate the potential of IM in the treatment of HPV-induced tumors.

## 4. Materials and Methods

### 4.1. Dermoscopic and In Vivo Confocal Laser Microscopy Evaluation of AGW

A 24-year-old patient with a genital wart on the urethral orifice was treated with a single application of IM 0.015% gel. IM was applied as an individual therapeutic attempt, after several surgical treatments as well as topical treatment with imiquimod, podophyllotoxin, and sinecatechins had failed. The patient had given informed consent to the treatment and the publication of photographs. Before and 8 h after treatment, the wart was assessed dermoscopically and by in vivo confocal laser scanning microscopy (Viva Scope 1500, Viva Scope GmbH, Munich, Germany).

### 4.2. Biopsies of IM-Treated Anogenital Warts

Patients with multiple AGW in the genital area, who received treatment with IM 0.05% gel (one single application) at the Department of Dermatology, Venerology and Oncodermatology, University of Pécs, Hungary, were asked whether warts could be superficially removed during treatment. Seven patients agreed and were recruited. The patients’ clinical data are summarized in Table 2. The sample collection was approved by the local ethics committee of the University of Pécs (# 6954-PTE2018). After informed consent, warts were obtained by shave biopsy at three different time points: before the treatment (t0) as well as 3 (t3) and 8 (t8) h after IM treatment. The warts were immediately fixed in 4% formalin solution and in an optimal cutting temperature (OCT) compound (Sakura Finetek USA, USA). HPV infection with subtype was confirmed on paraffin embedded warts (t0) by short-fragment group-specific HPV-PCR followed by HPV-typing with a reverse line-blot assay that covers 32 alpha-HPV types (INNO-LiPA HPV Genotyping Extra II, Fujirebio, Gent, Belgium), as previously described in more detail [55].

### 4.3. Histomorphology, Immunohistochemistry, and Immunofluorescence

Tissue sections (3 µm) were cut from formalin-fixed and paraffin-embedded tissue blocks and stained with hematoxylin and eosin (H&E). For immunohistochemistry, tissue was deparaffinized and rehydrated with distilled water. Subsequently, ‘heat-induced epitope retrieval’ was performed with respective buffers (CD4, CD8, Myeloperoxidase (MPO): EDTA buffer pH 9.1 (DCS, Hamburg, Germany), CD68: citrate buffer pH 6.1 (DCS, Germany)). The following antibodies were used in respective dilutions: CD4 (clone 4B12; Agilent, Santa Clara, CA, USA; 1:50 in DCS dilution buffer); CD8 (clone C8/144B; Agilent, USA; 1:100 in DCS dilution buffer); CD68 (clone KP1; Agilent, USA; 1:400 in DCS dilution buffer); and MPO (polyclonal; Agilent, USA; 1:20,000 in Agilent dilution buffer). Counterstaining was performed with hematoxylin after Mayer and slides were mounted with Aquatex (Merck, Darmstadt, Germany). Histomorphology was systematically evaluated using a list of histologic criteria for toxic changes of the epidermis, vasculitis, and livedoid vasculopathy (Table 2). The involvement of inflammatory cells was assessed on immunohistochemical stained slides (MPO: neutrophils, CD4: T helper cells, CD8: cytotoxic T cells, CD68: tissue macrophages). The presence or absence of histopathological changes was documented with a semi-quantitative score: o = not present, + = present, ++ = moderately present, +++ = strongly present, x = not assessable.

For immunofluorescence of neutrophils extracellular traps (NETs), tissue samples on glass coverslips were deparaffinized as described above and treated with 1% BSA to block unspecific antibody binding. Slides were stained with polyclonal anti-human MPO (IgG, sheep, 1:250, Antibody Online #AA16-718) and anti-mouse/human H3cit (IgG, rabbit, 1:500, Abcam, Cambridge, UK, #ab5103) as primary antibodies and polyclonal anti-sheep Alexa 568 (IgG, donkey, 1:500, Abcam, UK, #ab175712) and polyclonal anti-rabbit Alexa Fluor 488 (IgG, goat, 1:500, Invitrogen, Waltham, MA, USA, #A-11034) as secondary antibodies. DNA was stained with Hoechst (Sigma-Aldrich, Darmstadt, Germany). Confocal microscopy was performed on an Olympus IX83 inverted microscope (software: Olympus Fluoview Ver.4.2, Olympus) at 20× and 60× magnification, as indicated. Hoechst fluorescence was detected at 405 nm, H3cit fluorescence at 488 nm, and MPO fluorescence at 568 nm.

For immunofluorescence analysis of von Willebrand factor (VWF) and neutrophils on cryosections (10 µm), the following primary antibodies were used after methanol fixation: rabbit anti-human VWF (1:150; Agilent, USA), rat anti-human CD15 (1:150 (IgM; Abcam, UK)), and mouse anti-human CD31 (1:50; Agilent, USA). The following secondary antibodies were used: FITC-conjugated goat anti-rabbit IgG (1:200; BD Pharmingen, San Diego, CA, USA), Alexa 555-conjugated goat anti-rat IgG (1:200; Invitrogen, USA), Alexa 555-conjugated goat anti-rat IgM (1:500; Invitrogen, USA), and Alexa 555-conjugated goat anti-mouse IgG (1:200; Invitrogen, USA). Nuclei were counterstained with DAPI (1:1000). Fluorescence images were acquired with a Zeiss Axiovert 200 microscope and were analyzed using AxioVision v4.8 (Zeiss) and ImageJ v1.47c.

### 4.4. Cell Culture

Human umbilical vein endothelial cells (HUVECs) were obtained and maintained as described before [56]. Endothelial cells were grown to confluence in gelatin-covered 12-well plates. After being rinsed twice with HBRS, endothelial cells were stimulated with IM (10 nM, IM was provided by Leo Pharma, Ballerup, Denmark). Thrombin (0.5 U/mL) was used as positive control, and HEPES-buffered Ringer’s solution (HBRS) served as negative control. After 15 min, endothelial cell supernatant was harvested, centrifuged, and stored at −20 °C for subsequent ELISA experiments. HUVECs cultivated on coverslips were used for immunofluorescence studies. After fixation with ice-cold methanol for 5 min, the following primary antibodies were used: rabbit anti-human VWF (1:150; Agilent, USA) and mouse anti-human CD31 (1:50; Agilent, USA). The following secondary antibodies were used: FITC-conjugated goat anti-rabbit IgG (1:200; BD Pharmingen, USA) and Alexa 555-conjugated goat anti-mouse IgG (1:200; Invitrogen, USA).

The aneuploid immortal keratinocyte cell line from adult human skin (HaCaT) was cultured in calcium-free DMEM (HyClone #SH3031901), supplemented with 10% standardized foetal bovine serum (FBS Advance; Capricorne), 2 mM l-glutamine, 100 U/mL penicillin, and 0.1 mg/mminL streptomycin. HaCaTs were cultured in a humidified incubator at 37 °C and 5% CO_2_. IM (10 µM, IM was provided by Leo Pharma, Denmark) was solubilized in DMSO. Cells were treated with 10 nM of IM or with the solvent DMSO for 24 h. 

### 4.5. ELISA 

VWF secretion in the supernatant was quantified by ELISA as previously described [41]. Briefly, endothelial cells were stimulated with HBRS, Thrombin (0.5 IU/mL), or IM (10 nM), and 100 μL of the respective supernatants (sample or standards) was transferred to an anti-vWF pre-coated 96-well plate and incubated for 2 h at room temperature (RT). Afterwards, the plate was washed twice with 400 µL wash buffer (0.05% Tween^®^ 20 in PBS, pH 7.2–7.4) to remove unbound vWF, incubated with 100 μL of the detection antibody for 2 h at RT, followed by a repeated wash step, and incubation with 100 μL of the working dilution of Streptavidin-HRP for 20 min at RT. For sufficient detection of vWF, 100 μL of substrate solution was added to each well (20 min at RT) and the reaction was then blocked by adding 50 μL of stop solution. Differences in optical density of each well was determined by using a microplate reader at 450 nm including wavelength correction at 540 nm.

### 4.6. Transfection

Transfections with Lipofectamine 2000 (ThermoFisher Scientific, Waltham, MA, USA) were performed according to the manufacturer’s protocol. Briefly, liposome and the plasmid pLXSN16E6E7, or the control plasmid pLXSN, were incubated in DMEM without Ca^2+^ and FBS. The liposome-DNA mixture was added to each well (6-well plate, Sigma-Aldrich, Germany) containing HaCaT cells for 6 h, followed by a replacement with fresh culture medium for 24 h prior to treatment. 

### 4.7. RT-qPCR

Total RNA was isolated with the RNeasy Mini Kit (Qiagen, Hilden, Germany) according to the manufacturer’s instructions, and 1 µg was converted into cDNA using the High-Capacity cDNA Reverse Transcription Kit and random primers (ThermoFisher Scientific, USA). IL8 (CXCL8) expression in HaCaT cells was analyzed by SYBR green quantitative PCR (qPCR) (Platinum SYBR Green qPCR SuperMix-UDG w/Rox; ThermoFisher Scientific, USA) using CXCL8 Primer (IL-fw: 5′-AGACAGCAGAGCACA-3′, IL-8rv: 5′-ATGGTTCCTTCCGGTGGT-3′), with the expression values normalized to the house-keeping genes glyceraldehyde-3-phosphate dehydrogenase (*GAPDH*_fw: 5′-GCAAATTCCATGGCACCGT-3′, GAPDH_rv: 5′-GCCCCACTTGATTTGGAGG-3′) and Hs_ACTB_1_SG QuantiTect Primer QIAGEN (Cat. No. QT00095431). For all qPCR setups, samples from independent experiments were run in triplicates. For statistical analysis, the delta-delta-Ct method was used. In brief, Ct values for genes of interest in the individual samples were normalized to the housekeeping genes GAPDH (Glyceraldehyde-3-Phosphate Dehydrogenase) and ACTB (beta-Actin), resulting in ΔCt values. Mean ΔCt of the control samples was calculated, and ΔΔCt values of all individual samples were calculated as the difference between the ΔCt value of the individual sample and the mean control ΔCt. 2-ΔΔCt was used to calculate the relative fold gene expression levels. Two-way ANOVA on ΔΔCt values was used to analyze the statistical significance of the differences.

## Figures and Tables

**Figure 1 ijms-23-13377-f001:**
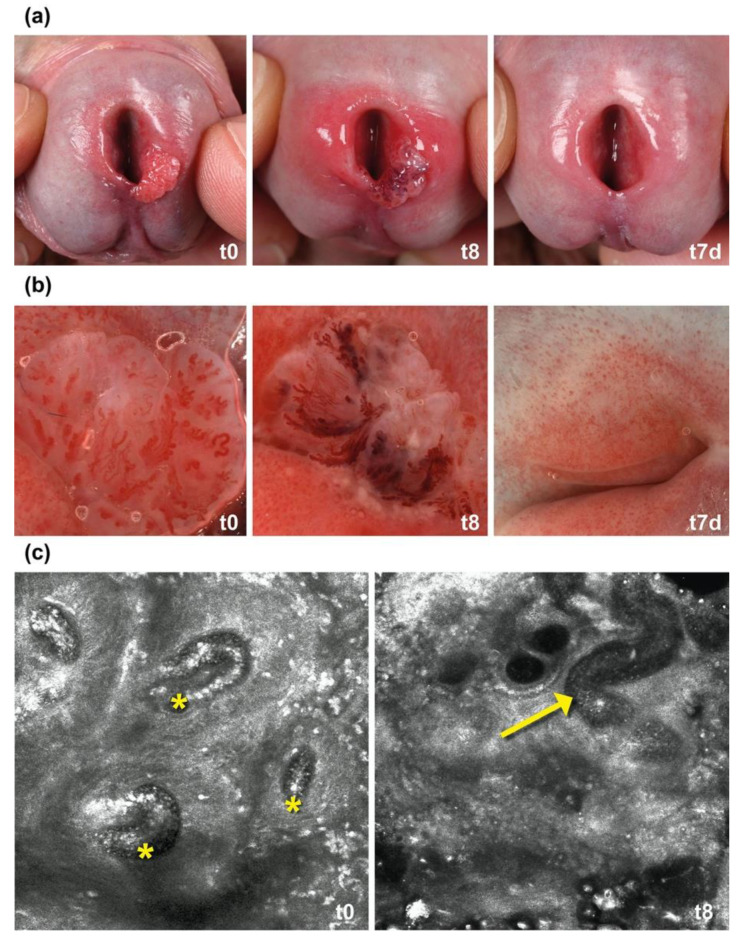
(**a**) Ingenol mebutate (IM) gel induces fast ablation of anogenital warts (AGW). One drop of IM 0.015% gel was applied on a single wart at the urethral orifice. After 8 h the wart showed clinical signs of necrosis (t8) and peeled of within the next 24 h. After one week the treatment site had completely healed (t7d); (**b**) dermoscopic changes during the treatment. After 8 h the wart’s vessels partly showed a livid erythematous color with distinct caliber variations (t8); (**c**) in vivo confocal laser microscopy videos reveal thrombosis in small vessels. Before the treatment, capillaries in the papillary dermis showed a regular blood flow (t0, yellow stars). After 8 h the blood flow in the capillaries stopped in the sense of microthrombosis (yellow arrow). The videos are provided in the Appendix A.

**Figure 2 ijms-23-13377-f002:**
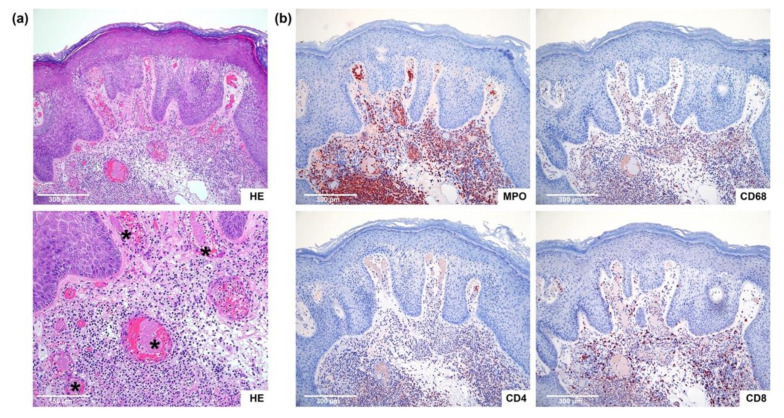
(**a**) Histology confirms microthrombosis in warts 8 h after treatment with ingenol mebutate (IM). The wart of patient 6 is shown as an example. Both within the capillaries and the postcapillary venules fibrinoid thrombi (*) could be detected, surrounded by sludge of erythrocytes. The postcapillary venules showed a strong diapedesis of neutrophils with extravasated erythrocytes and some distinct leukocytoclasia around the vessels, proving signs for vasculitis were missing. (Hematoxylin and Eosin (HE) staining; original magnification top 100×, bottom 200×); (**b**) immunohistochemistry showed a neutrophil-dominated inflammatory infiltrate after 8 h. Again, patient 6′s wart is shown as an example at timepoint 8 h. The inflammatory infiltrate consisted mainly of MPO-positive neutrophils, but also CD68-positive macrophages, CD8-positive cytotoxic T cells, and CD4-positive T helper cells were recruited. (Immunohistochemical staining with MPO, CD68, CD4, CD8; original magnification 100×).

**Figure 3 ijms-23-13377-f003:**
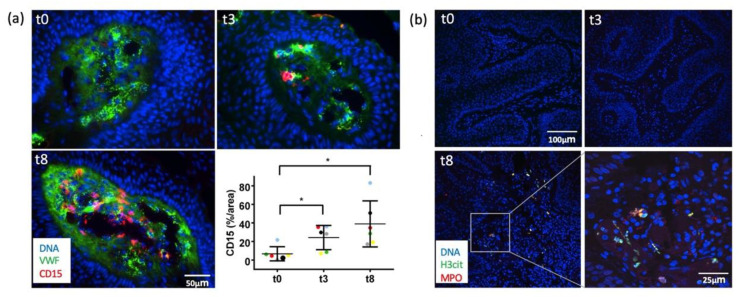
(**a**) Ingenol mebutate (IM) mediates von Willebrand factor (VWF) secretion, neutrophil infiltration, and NET formation. Tissue sections were stained for VWF (green), and CD15 for neutrophils (red). Nuclei are stained with DAPI (blue). After topic administration of IM, VWF fibers are detectable within the vessel lumen indicating endothelial cell activation. These VWF networks are associated with a massive increase of luminal neutrophils over time (please see for quantification the graphical inset, the different colored dots indicate warts of different patients). Results are expressed as relative area (%) of the vessel lumen and data are presented as mean ± SD of each patient. Tissue of six patients (n = 6) was analyzed (* *p*  ≤  0.05); (**b**) tissue sections of all six patients (n = 6) were stained for H3cit (green) as a marker of NET formation and MPO (red) as a marker for neutrophils at the indicated time-points. Chromatin was stained with Hoechst (blue). After 8 h (t8), neutrophil infiltration and NET formation was visible. Exemplary presentation of patient 6. The evaluations of additional patients can be found in the Appendix A.

**Figure 4 ijms-23-13377-f004:**
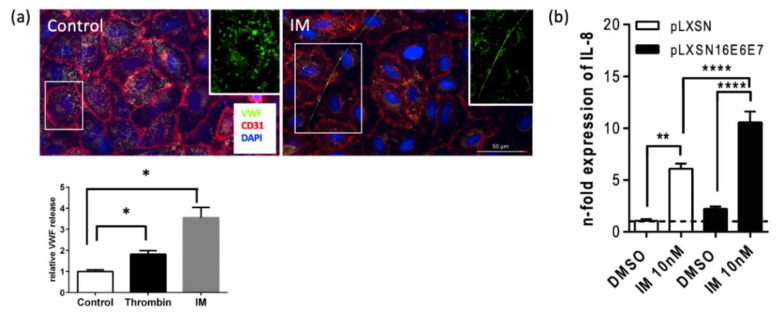
(**a**) Ingenol mebutate (IM) mediates endothelial cell activation and von Willebrand factor (VWF) secretion. A HUVEC monolayer was stained for VWF (green) and CD31 (red). Nuclei are stained with DAPI (blue). Representative images of quiescent endothelial cells show intracellular stored VWF (upper panel, left). Incubation with IM triggers the release of VWF and the formation of luminal VWF strings (upper panel, right). Activation of endothelial cells was induced by thrombin or IM and the release of VWF was quantified in the supernatant (lower panel, left). Bar graphs represent relative VWF release ± SD of two independent experiments (n = 4) (* *p* ≤ 0.05); (**b**) expression of IL-8 in E6/E7-expressing HaCaTs upon treatment with IM. HaCaT cells were transfected with pLXSN16E6E7 or the control plasmid pLXSN. After 48 h, cells were treated with IM (10 nM) or the solvent DMSO and IL-8 expression was analyzed via qPCR. Bar graphs represent relative expression levels ± SEM relative to the two reference genes GAPDH, actin β (ACTB) of five independent experiments (n = 5). Two-way ANOVA followed by Tukey’s multiple comparisons test. ** *p* ≤  0.01, **** *p* ≤ 0.0001.

**Table 1 ijms-23-13377-t001:** Histomorphological and immunohistochemical assessment.

	Patient #1			Patient #2			Patient #3			Patient #4			Patient #5			Patient #6			Patient #7		
Timepoint [h]	0	3	8	0	3	8	0	3	8	0	3	8	0	3	8	0	3	8	0	3	8
Hyperkeratosis	o	o	+++	+	+	+	+	+	+	+	++	+	+	+	o	++	++	++	++	+	+
Parakeratosis	o	o	o	o	o	o	o	o	o	o	+	o	o	o	o	o	+	+	+	+	+
Spongiosis	o	o	+	o	o	o	o	o	++	o	o	++	o	o	o	o	o	+	o	+	+
Pallor of the epidermis	o	o	+	+	++	++	o	o	++	o	+	++	o	+	++	o	+	+	o	+	+
Necrotic, apoptotic keratinocytes	o	o	o	o	+	+	o	o	+	o	o	+	o	o	+	o	+	+	o	+	+
Ballooning degeneration	o	o	o	+	o	++	o	o	++	o	o	++	o	o	+++	o	o	+	o	+	+
Reticulated degeneration	o	o	o	o	o		o	o	o	o	o	++	o	o	+++	o	o	+	o	o	+
Extravasated erythrocytes around capillaries	o	o	+	o	+	++	o	o	+++	o	o	+++	o	o	++	o	o	+	o	o	o
Thrombus in capillaries	o	o	o	o	o	o	o	o	+	o	o	+	o	o	+	o	o	+	o	+	+
Thrombus in postcapillary venules	o	o	o	o	o	x	o	x	+	o	o	+	o	o	x	o	o	+	o	x	x
Fibrinoid degeneration of vessel walls	o	o	o	o	o	o	o	o	o	o	o	o	o	o	o	o	o	o	o	o	o
Fibrin perivascular	o	o	o	o	o	o	o	o	o	o	o	o	o	o	o	o	o	+	o	o	o
Extravasated erythrocytes around postcapillary venules	o	o	o	o	o	x	o	x	++	o	o	o	o	+	x	o	o	+	o	x	x
Diapedesis of neutrophils	o	o	o	o	o	x	o	x	++	o	o	+	o	o	x	o	o	+++	o	x	x
Leukocytoclasia	o	o	o	o	o	+	o	o	++	o	o	o	o	o	+	o	o	+	o	o	o
Intraepidermal neutrophils (MPO)	o	o	o	o	o	o	o	o	+	o	o	o	o	o	+	o	o	+	o	o	+
Intravascular neutrophils (MPO)	o	o	+	+	++	++	o	+	+++	o	o	+++	o	++	++	o	+	+++	o	++	+++
Perivascular neutrophils (MPO)	o	+	+	o	++	++	o	+	+++	o	+	+++	+	++	++	o	+	+++	+	++	+++
Diapedesis neutrophils (MPO)	o	+	o	o	+	+	o	o	++	o	o	+++	o	+	++	o	o	+++	0	+	+++
Dermal T helper cells (CD4)	+	+	++	+	+	+	+	+	+	+	+	++	++	++	+	++	++	++	+	++	++
Dermal cytotoxic T cells (CD8)	+	+	++	+	+	+	+	+	+++	+	+	++	++	++	+	++	++	+++	+	++	+++
Dermal macrophages (CD163)	+	+	+	+	+	+	+	+	+	+	+	+	+	+	+	++	++	++	+	++	++

Semi quantitative score: o = not present, + = present, ++ = moderately present, +++ = strongly present, x = not assessable.

**Table 2 ijms-23-13377-t002:** Clinical data of the patients with genital warts.

	Patient #1	Patient #2	Patient #3	Patient #4	Patient #5	Patient #6	Patient #7
Gender	M	W	M	M	M	M	M
Age	27	22	22	28	50	62	64
History of warts	24 months	7 months	6 months	10 months	12 months	2 months	12 months
Location	Mons pubis, penile shaft	Labia maiora	Mons pubis	Mons pubis, penile shaft	Penile shaft	Penile shaft	Penile shaft
Previous treatments	Cryotherapy Podophyllo-toxin	-	-	-	Cryotherapy	-	Cryotherapy
Comorbidities	-	-	Gastro-esophageal reflux disease	-	-	Arterial hypertension, peripheral artery disease	Arterial hypertension
HPV type	HPV 6	HPV 6	HPV 6	HPV 6	HPV 6	HPV 6	HPV 6

## Data Availability

The authors confirm that the data supporting the findings of this study are available within the article and its Appendix A. Raw data are available on request from the corresponding author, S.A.B.

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
