# Peer review of "Immunothrombotic Mechanisms Induced by Ingenol Mebutate Lead to Rapid Necrosis and Clearance of Anogenital Warts"

_ijms, 2022, doi:10.3390/ijms232113377_

Round 1

Reviewer 1 Report

This paper entitled “Immunothrombotic mechanisms induced by ingenol mebutate lead to rapid necrosis and clearance of anogenital warts” by Braun et al. investigated the mechanisms of ingenol mebutate treating anogenital warts.

HPV induced anogenital warts are a common therapeutic challenge. The effectiveness of IM managing genital warts in sensitive anatomic locations has been widely reported. This study tried to study the mechanisms, which remain unknown. The overall novelty is good.

The overall quality of this work is average. The manuscript is well-prepared. However, there are some remain questions with experiment design as follows:

Major concern:

1 Authors has showed IM mediates vWF secretion, which also been reported. But authors may need further proof the effectiveness of IM specificity on HPV induced anogenital warts is related to the vWF secretion

2 In Figure 2, There is no comparison between treatment and control. It is hard to tell any recruitment without proper control.

3 Authors only used HUVEC might not a good model as it is endothelial cells. Would the authors mind to explain why they pick HUVEC and HaCaT cells and switch between them?

4 P8 line 27 Authors claimed: “However, after 8 hours (t8), neutrophils and NETs were clearly visible within the papillary dermis, as exemplified in the immunofluorescence images of patient 6 (Figure 3b).” I guess the evidence is too weak to conclusion that with only 1 immunofluorescence photo. It will be nice if author can provide further quantitative evidence to support their conclusion.

4 Limitation of this study might be discussed.

Minor issues:

1 No scale bar on Fig 2, Fig 3 (b). Missing scale bar on Figure 4(a)

2 More detail needed in section 4.5 ELISA

3 It will be nice if authors can include n=? in the figure legend.

Reviewer 2 Report

Dear authors, I find the manuscript very interesting. However I have some comments to make:

1. In the IHC studies, why didn't you choose another specific marker for neutrophils?

2. In general I noticed that you always change the markers, not demonstrating their presence / absence in all the tests carried out

3.In section 2.4, use different cell lines, first HUVec and then HaCaT. Why change your cell line. In general I find this paragraph confusing. To be remodulated it.

4. HaCaT is a cell line of immortalizing keratinocytes, with mutated p53. Are you sure that this did not affect your results? To discuss it.

5. What genotypes do the 7 patients have for HPV? It would be better to specify it, since we are starting today with personalized therapyIt would be better to specify it, since we start today of personalized therapy

Reviewer 3 Report

The submitted study investigated the processes underlying ingenol mebutate's efficacy in the treatment of anogenital warts.

Major: 

Figure 3a. It is unclear which statistical method was used in this figure.

Figure 4b. All PCR amplicons must be amplified effectively for delta delta Ct analysis to be valid. Please include amplification efficiency in the Method or Rebuttal letter.

Figure 2, Figure 3. Please supply other sample photos (preferably from a different patient) as supplementary data.

Minor:

Future 3b has a typographical error. H4cit, should be H3cit.

Round 2

Reviewer 1 Report

I appreciate authors' effort on the response.  As authors have successfully answer all the questions, I recommend to published in present form.

Reviewer 2 Report

Authors modified the manuscript text and figures. I'm agree for pubblication.

Reviewer 3 Report

Thanks for the updates and congratulations on the great work.